# EM-RBR: A REINFORCED FRAMEWORK FOR KNOWLEDGE GRAPH COMPLETION FROM REASONING PERSPECTIVE

## ABSTRACT

Knowledge graph completion aims to predict the missing links among the knowledge graph (KG),i.e., predicting the possibility that a certain triple belongs to the knowledge graph. Most mainstream embedding methods focus on fact triplets contained in the given KG, however, ignoring the rich background information provided by logic rules driven from knowledge base implicitly. Limited to the modeling of algebraic space, contradictory in expressing certain relational patterns usually exists in the embedding models. Therefore, the representation of the knowledge graph is incomplete and inaccurate. To solve this problem, in this paper, we propose a general framework, named EM-RBR(*embedding and rule-based reasoning*), capable of combining the advantages of reasoning based on rules and the state-of-the-art models of embedding. EM-RBR aims to utilize relational background knowledge contained in rules to conduct multi-relation reasoning link prediction. In this way, we can find the most reasonable explanation for a given triplet to obtain higher prediction accuracy. In experiments, we demonstrate that EM-RBR achieves better performance compared with previous models on FB15k, WN18 and our new dataset FB15k-R, especially the new dataset where our model perform futher better than those state-of-the-arts. We make the implementation of EM-RBR available at `https://github.com/1173710224/link-prediction-with-rule-based-reasoning`.

## 1 INTRODUCTION

Knowledge graph (KG) has the ability to convey knowledge about the world and express the knowledge in a structured representation. The rich structured information provided by knowledge graphs has become extremely useful resources for many Artificial Intelligence related applications like query expansion (Graupmann et al., 2005), word sense disambiguation (Wasserman Pritsker et al., 2015), information extraction (Hoffmann et al., 2011), etc. A typical knowledge representation in KG is multi-relational data, stored in RDF format, e.g. *(Paris, Capital-Of, France)*. However, due to the discrete nature of the logic facts (Wang & Cohen, 2016), the knowledge contained in the KG is meant to be incomplete (Sadeghian et al., 2019). Consequently, knowledge graph completion(KGC) has received more and more attention, which attempts to predict whether a new triplet is likely to belong to the knowledge graph (KG) by leveraging existing triplets of the KG.

Currently, the popular embedding-based KGC methods aim at embedding entities and relations in knowledge graph to a low-dimensional latent feature space. The implicit relationships between entities can be inferred by comparing their representations in this vector space. These researchers (Bordes et al., 2013; Mikolov et al., 2013; Wang et al., 2014; Ji et al., 2015; Lin et al., 2015; Nguyen et al., 2017) make their own contributions for more reasonable and competent embedding. But the overall effect is highly correlated with the density of the knowledge graph. Because embedding method always fails to predict weak and hidden relations which a low frequency. The embedding will converge to a solution that is not suitable for triplets owned weak relations, since the training set for embedding cannot contain all factual triplets.However, reasoning over the hidden relations can covert the testing target to a easier one. For example, there is an existing triplet *(Paul, Leader-Of, SoccerTeam)* and a rule *Leader-Of(x,y) $\implies$ Member-Of(x,y)* which indicates the leader of a soccer

team is also a member of a sport team. Then we can apply the rule on the triplet to obtain a new triplet *(Paul, Member-of, SportTeam)* even if the relation *Member-of* is weak in knowledge graph.

Besides, some innovative models try to harness rules for better prediction. Joint models (Rocktäschel et al., 2015; Wang et al., 2019; Guo et al., 2016) utilize the rules in loss functions of translation models and get a better embedding representation of entities and relations. An optimization based on ProPPR (Wang & Cohen, 2016) embeds rules and then uses those embedding results to calculate the hyper-parameters of ProPPR. These efforts all end up on getting better embedding from rules and triplets, rather than solving completion through real rule-based reasoning, which is necessary to address weak relation prediction as mentioned before. Compared with them, EM-RBR can perform completion from the reasoning perspective.

Usually, there are some contradictions in the mathematical space of existing embedding-based models. Take transE (Bordes et al., 2013) and RotatE (Sun et al., 2019) as examples. For the relation pattern $R(x, y) \Rightarrow R(y, x)$, i.e., a symmetrical relation, transE cannot model it as described in Sun et al. (2019). While RotatE can not model some relation pattern. For example, transitive relation $R$, w.r.t $R(x, y) \wedge R(y, z) \Rightarrow R(x, z)$. We assume that the embedding of the relation $R$ under RotatE is $e^{i\theta_R}$, abbreviated as $r$. Formula 1 is necessary and sufficient to transitivity. So we will get $r^2 = r$, which means that $e^{i\theta_R} = 1$, so $\theta_R = 0, \theta_R \in [0, 2\pi)$. $\theta_R = 0$ denotes that $R$ is a reflexive relation. That is all the embedding of the transitive relation will be trained to have the nature of the reflexive relation, which is out of our expect.

$$x \cdot r = y, y \cdot r = z, x \cdot r = z \tag{1}$$

In addition to the above problems, transE and RotatE models cannot model data that has multiple relations between two entities. TransR (Lin et al., 2015) can solve this problem by training the relationship as a transformation matrix. But none of them can solve the problem that an entity has the same relation with multiple other entities. For example, there are two triples $(h, r, t_1), (h, r, t_2)$ in the knowledge graph($t_1 \neq t_2$). For the transE model, $\boldsymbol{h} + \boldsymbol{r}$ is a fixed result. So a wrong equation $\boldsymbol{r_1} = \boldsymbol{r_2}$ will exist under transE model. The other two mathematical models are no exception.

The above-mentioned shortcomings can be solved by defining the relation pattern in a rule directly. Morivated by this, We propose a novel framework EM-RBR combing embedding and rule-based reasoning, which is a heuristic search essentially. In the development of the joint framework EM-RBR, we meet two challenges. On the one hand, we use AMIE (Galárraga et al., 2013) to auto-mine large amount of rules but not manually. However, these rules automatically mined sometimes are not completely credible. Therefore, it is necessary to propose a reasonable way to measure rules to pick proper rules when reasoning. On the other hand, it is known that traditional reasoning-based methods will give only 0 or 1 to one triplet to indicate acceptance or rejection for the given knowledge graph. This conventional qualitative analysis lacks the quantitative information as the embedding models. So the result of EM-RBR need to reflect the probability one triplet belonging to the knowledge graph.

Three main contributions in EM-RBR are summarized as follows:

- EM-RBR is flexible and general enough to be combined with a lot of embedding models.
- We propose a novel reasoning algorithm, which can distinguish a given triplet with other wrong triplets better.
- We propose a novel rating mechanism for auto-mined reasoning rules and each rule will be measured properly in our framework.

In the remaining of this paper, we will explain how our model works in Section 2, experiments in Section 3 and related work in Section 4.

## 2 METHOD

Our motivation is to use rules to make up for the shortcomings of embedding model. When the mathematical space of a certain embedding model is not enough to represent a certain relation pattern, the relation pattern can be declared through rules. For example, for the symmetric relationship $R$, we declare a rule $R(x, y) \Rightarrow R(y, x)$. When scoring a certain triple $(a, R, b)$ under transE model, we can take $\min(s(a, R, b), s(b, R, a))$ as the The final score of the triple. This is equivalent

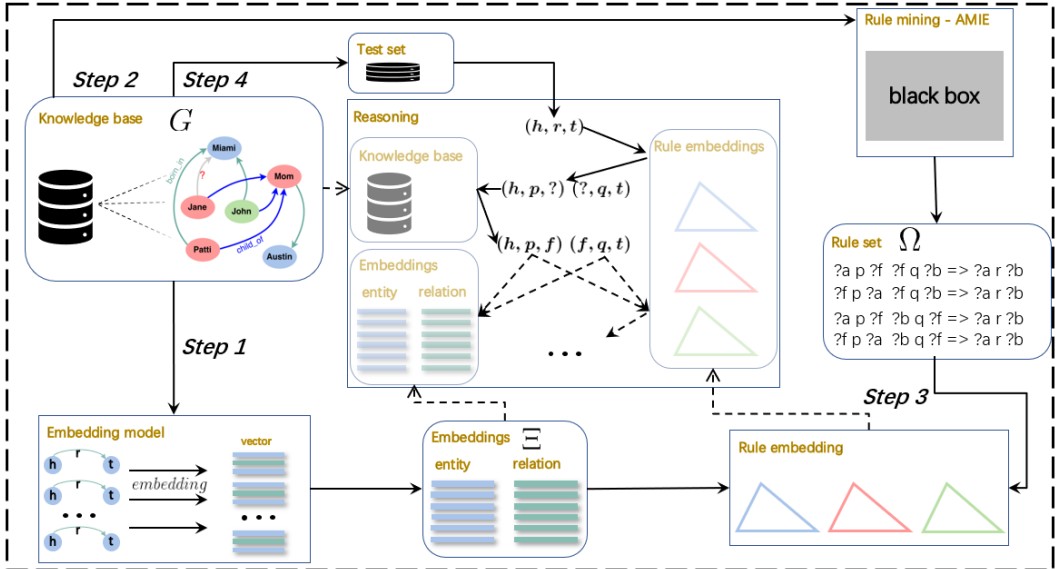

Figure 1: An overview of our framework.

to finding another triple that can explain the rationality of this triple through rules. Although the relation between the new triplet and the original triplet cannot be expressed in the algebraic space of embedding, the most reasonable explanation can be found through rules to solve the defects of the model.

The core idea of our framework is to conduct multi-relation path prediction in deeper context from reasoning perspective, that is in the form of heuristic search. Before explaining the concrete reasoning algorithm, let's take an overview of our framework in Section 2.1.

## 2.1 OVERVIEW

**Definition 2.1. *Rule:*** *A rule in our framework is in the form of* $B_1(x, z) \land B_2(z, y) \implies H(x, y)$ *or* $B(x, y) \implies H(x, y)$.

We model a knowledge graph as a collection of facts $G = \{(h, r, t) | h, t \in \mathcal{E}, r \in \mathcal{R}\}$, where $\mathcal{E}$ and $\mathcal{R}$ represent the set of entities and relations in the knowledge graph, respectively. The steps of our framework are as follows corresponding to Figure 1.

> ***Step 1***. We invoke an embedding model to get a set $\Xi \in \mathbb{R}^{(|\mathcal{E}|+|\mathcal{R}|) \times k}$ containing the $k$-dimensional embedding of entities and relations in $G$.

> ***Step 2***. We apply AMIE (Galárraga et al., 2013) on $G$ to get the reasoning rule set $\Omega$, where each rule meets Definition 2.1.

> ***Step 3***. The reasoning rules are measured based on the embedding of relations contained in the rule, which will explained in Secion 2.2.2.

> ***Step 4***. Reasoning is conducted for a given triplet $(h, r, t)$, which will be described in Section 2.2.

## 2.2 REASONING ALGORITHM

**Definition 2.2. *initial state:*** *Initial state is represented as a set consisting of a single triplet, i.e.* $\{(h, r, t)\}$. ***state:*** *According to the initial state and the state's expansion method described in Section 2.2.1, each state is represented as a set a triples.*

This reasoning algorithm is the core of our framework and it's a heuristic search. The input of this algorithm is a triplet $(h, r, t)$. State in the search space is as Definition 2.2. Each state has two

scores, i.e. $\mathcal{H}$ and $\mathcal{L}$. $\mathcal{H}$ is the heuristic score of state and $\mathcal{L}$ is the score of the state. The calculation of scores is explained in Section 2.2.2.

The target of search is to compute $\Phi_{\sim(h,r,t)} = \min_{state \in \mathbb{S}} \{\mathcal{L}_{\sim state}(h,r,t)\}$ for a specific triple $(h,r,t)$. The score $\Phi$ of a triplet meets $\Phi \geq 1$. The smaller $\Phi$ is, the triplet belongs to knowledge graph with greater probability.

The pseudo code of search is as shown in Appendix B. We use a priority queue $Q$ to store states in ascending order of $\mathcal{H}$. The initial state is the target triplet itself, whose $\mathcal{H}$ is 1. Push the initial state into $Q$ and then begin the loop until $Q$ is empty.

During the search process, pop the top of $Q$ as the current state $s_{cur}$. It will not be extended if $\mathcal{H}_{s_{cur}} \geq \Phi$, otherwise we will extend it by matching rules to get new states. For each new state $s_{new}$, compute its score $\mathcal{H}_{s_{new}}$ and $\mathcal{L}_{s_{new}}$. If $\mathcal{H}_{s_{new}} < \mathcal{L}_{s_{cur}}$, the state will then be pushed into $Q$.

### 2.2.1 MATCHING AND EXTENSION

State is a set of triplets, the initial state is the target triplet itself. Intermediate states are extended from the initial state. So essentially, the extension of a state is extension of the triplets in the state. For a triplet $(h,r,t)$, the process of matching and extension is roughly as follows:

 *1.* Find rules $\omega \in \Omega$ in the form of $B_1(x,z) \wedge B_2(z,y) \implies H(x,y)$[1], where $H = r$.

 *2.* Assign entities to variables in the rule, i.e. $x = h, y = t$.

 *3.* Find all $z_0$ that satisfy $(x, B_1, z_0) \in G$ or $(z_0, B_2, y) \in G$, where $x = h, y = t$.

 *4.* $(h,r,t)$ is extended to $\{(h, B_1, z_0), (z_0, B_2, t)\}$. A triplet always has multiple extensions.

For example, we expand the target triplet in the initial state. There are two triplets in the sub-state, and either of them must be in the knowledge graph. When the sub-state is further expanded, the triplet in the knowledge graph need not to be expanded. Therefore, there should be $m + 1$ triplets in each sub-state after extending $m$ times. And at least $m$ of them belong to the knowledge graph.

### 2.2.2 COMPUTATION OF $\mathcal{H}$ AND $\mathcal{L}$

$\mathcal{H}_{\sim \mathcal{O}}(h,r,t)$ denotes the heuristic score of triplet $(h,r,t)$ when extended to state $\mathcal{O}$ and $\mathcal{L}_{\sim \mathcal{O}}(h,r,t)$ is the corresponding state score.

$$\mathcal{H}_{\sim \mathcal{O}}(h,r,t) = \prod_{(B_1 \wedge B_2 \Rightarrow H) \in \Delta_{Path}} \omega(B_1, B_2, H) \quad \textbf{\textit{w.r.t.}} \; \omega(B_1, B_2, H) \leftarrow e^{\frac{||\boldsymbol{B_1} + \boldsymbol{B_2} - \boldsymbol{H}||}{k}} \quad (2)$$

$\mathcal{H}_{\sim \mathcal{O}}(h,r,t)$ is defined as Equation 2 indicating the product of the scores of all the rules. $\Delta_{Path}$ represents the set of the rules used in the extension from the initial state to the current state. $\omega(B_1, B_2, H)$ is the score of rules in the shape of $B_1 \wedge B_2 \Rightarrow H$.

$$\mathcal{L}_{\sim \mathcal{O}}(h,r,t) = \mathcal{H}_{\sim \mathcal{O}}(h,r,t) * \prod_{(\mathcal{O}_h, \mathcal{O}_r, \mathcal{O}_t) \in \mathcal{O}} s_{\sim transX}(\mathcal{O}_h, \mathcal{O}_r, \mathcal{O}_t) \quad (3)$$

$\mathcal{L}_{\sim \mathcal{O}}(h,r,t)$ is defined as Equation 3 indicating the product of $\mathcal{H}_{\sim \mathcal{O}}(h,r,t)$ and the scores of all the triplets in the state. $\mathcal{O}$ denotes the state and $(\mathcal{O}_h, \mathcal{O}_r, \mathcal{O}_t)$ is a triplet belongs to $\mathcal{O}$. $s_{\sim transX}(\mathcal{O}_h, \mathcal{O}_r, \mathcal{O}_t)$ is the embedding score of this triplet as defined in Equation 4.

$$s_{\sim transX}(\mathcal{O}_h, \mathcal{O}_r, \mathcal{O}_t) = \begin{cases} 1 & if \; (\mathcal{O}_h, \mathcal{O}_r, \mathcal{O}_t) \in G \\ ||\boldsymbol{O_h} + \boldsymbol{O_r} - \boldsymbol{O_t}||/k + 1 & if \; (\mathcal{O}_h, \mathcal{O}_r, \mathcal{O}_t) \notin G \end{cases} \quad (4)$$

*Rule's score*
To evaluate the score of rule $B_1(x,z) \wedge B_2(z,y) \implies H(x,y)$, we visualize the three triplets of this rule in a two-dimensional space in Figure 3 of Appendix A. In our model, if a rule has a high confidence, it should satisfy $||x + H - y|| \approx ||x + B_1 - z + z + B_2 - y||$. We have $\boldsymbol{H} \approx \boldsymbol{B_1} + \boldsymbol{B_2}$, so we can use $||\boldsymbol{B_1} + \boldsymbol{B_2} - \boldsymbol{H}||$ to evaluate the score of the rule. $k$ is the dimension of embedding.

---

[1]The rules we analyzed here are in the form of $B_1(x,z) \wedge B_2(z,y) \implies H(x,y)$. As for rules like $B(x,y) \implies H(x,y)$, the process is similar and will not be overtalked here.

The smaller score, the higher confidence. To make the dimension in the calculation uniform, we divide the score of the rule by $k$. And then perform the $e$ exponential transformation to get the form in the Equation 2. The reason for this transformation will be explained in section 2.4.

*Triplet's score*
$||\mathcal{O}_h + \mathcal{O}_r - \mathcal{O}_t||$ is the score of triplet $(\mathcal{O}_h, \mathcal{O}_r, \mathcal{O}_t)$ in transE model[2]. The smaller the value, the more likely the triplet is in $G$. When $(\mathcal{O}_h, \mathcal{O}_r, \mathcal{O}_t) \in G$, the score is assumed to be 0. The same to rule's score, we also perform a certain transformation on the scores of the triplets, which is to divide by $k$ and add 1.

## 2.3 EXAMPLE

**Assumption 2.1.** *We put all the necessary message in Table 3 and 4. Apart from that, we make two assumptions. One is that we use the same symbol $r_i$ to represent a rule's symbol and rule's score. Another is that we define some data relations as Equation 5.*

$$r_1 r_3 r_5 > \mathcal{L}_{\sim s_3}(h, r, t) \ \& \ \mathcal{H}_{\sim s_7}(h, r, t) > \Phi_{\sim (h, r, t)} \tag{5}$$

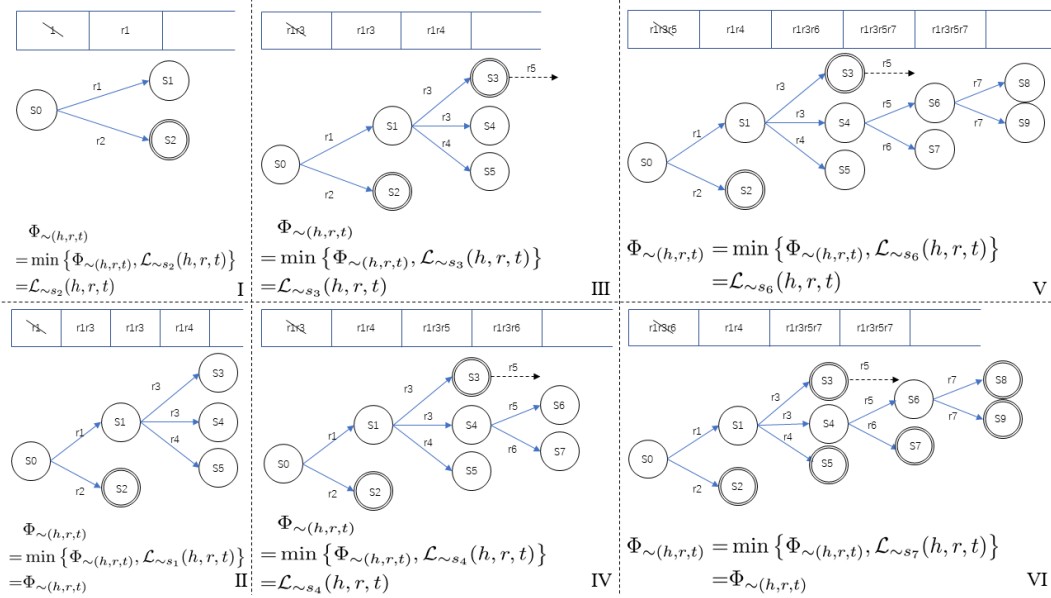

Figure 2: Demonstration of the search process based on an example. The search process is divided into six stages, each stage is contained in a sub-graph, each sub-graph contains three parts. The top of the sub-graph shows the current state of the priority queue, the middle part is the visualization of the search, the formula for updating $\Phi_{\sim (h, r, t)}$ at each stage is given at the bottom.

In this section, we use an example to illustrate our algorithm as shown in Figure 2[3]. The initial state $s_0$ only contains one triplet $(h, r, t)$, and its state score and heuristic score are both 1. At the beginning, the priority queue $Q$ has only one element, i.e. the initial state with its scores. The search process is as follows, and the necessary message is defined in Assumption 2.1.

I. $s_0$ matches $r_1$ and $r_2$ and extends to $s_1$ and $s_2$ respectively. $s_2$ is a termination state for the triplets in $s_2$ are all in $G$. We use $\mathcal{L}_{\sim s_2}(h, r, t)$ to update $\Phi_{\sim (h, r, t)}$ and push $s_1$ into $Q$.

II. Pop the top of queue $s_1$. Use it to update $\Phi_{\sim (h, r, t)}$ and then extend it to three new states which will be pushed to $Q$.

---

[2]Here we take transE as an example, so we use $||\mathcal{O}_h + \mathcal{O}_r - \mathcal{O}_t||$. If the combined model changes, this formula should change to the form in the combined model, too.

[3]There will be some conflicts in the usage of symbols. For these symbols, it's only valid in this example.

III. Pop the top of queue $s_3$ to update $\Phi_{\sim(h,r,t)}$ and extend it with matching the rule $r_5$. Since $r_1 r_3 r_5 > \mathcal{L}_{\sim s_3}(h,r,t)$, i.e. the solution produced by this path will not be the global minimum. As a consequence, this state is no longer extended.

IV. Pop the top of queue $s_4$ to update $\Phi_{\sim(h,r,t)}$ and extend to get two new states $s_6, s_7$.

V. Pop the top of queue $s_6$ to update $\Phi_{\sim(h,r,t)}$ and extend to $s_8, s_9$ after the rule $r_7$.

VI. Pop the top of queue $s_7$ and now $\mathcal{H}_{\sim s_7}(h,r,t) > \Phi_{\sim(h,r,t)}$. So $s_7$ and the remaining states in $Q$ need not extend. Therefore, all the remaining states in $Q$ become termination states. The search stops.

## 2.4 ANALYSE

### IS THE ALGORITHM SURE TO BE EFFECTIVE?

For three real number $a, b, c (a, b, c > 1)$, it's possible that $c > a * b * 1$. Consider triplet $(h, r, t)$ and rule $B_1(x, z) \wedge B_2(z, y) \implies H(x, y)$ ,w.r.t $r = H$. If $c$ represents the score of $(h, r, t)$, $a$ represents the score of the rule, $b$ represents the score of the expanded new triplet that not in the knowledge graph, and $1$ represents the score of the expanded new triplet that in the knowledge graph. Then the score of $(h, r, t)$ will be reduced to $a * b$, i.e. we use the new expanded triplets and the rule to evaluate $(h, r, t)$.

Of course, this optimization will be effective only on the correct triplets. For the wrong triplets, another wrong triplet with a large score will be obtained after rule matching. So $c > a * b$ in this occasion is a very unlikely event. As a result, the correct triplets are optimized, and the wrong triplets will generally not be optimized. Therefore, from a macro perspective, the ranking of the correct triplets will increase.

### IS THIS ALGORITHM A TERMINATING ALGORITHM?

The heuristic score of a state is the product of scores of all the rules along the reasoning path. The scores of the rules are all number greater than 1, so when a search path is long enough, $\mathcal{H}$ must be greater than $\Phi$. So the search must be able to stop.

### WHAT IS THE PRINCIPLE WHEN DESIGNING THE CALCULATION OF RULES AND TRIPLETS?

From the above, we require that the scores of rules and triplets are greater than 1 and close to 1. Given that each dimension of embedding obtained by the translation model is a number less than 1 and close to 0, we divide the score by the corresponding dimension $k$ and then add 1 to meet our requirements. In addition, in order to highlight the importance of rules in the calculation, we use exponential changes for rules instead of plus 1.

## 3 EXPERIMENT

We would like to prove two things. One is that EM-RBR is a valid reinforced model, i.e. EM-RBR(X) model always performs better than X model. Another is that EM-RBR will beat all of the current state-of-the-arts on a data set with rich rules.

### 3.1 EXPERIMENT SETUP

**Dataset:** We evaluate EM-RBR on FB15k, WN18 (Bordes et al., 2013), FB15k237 (Toutanova & Chen, 2015) and WN18RR (Dettmers et al., 2017). We propose a new data set FB15k-R, whose test set contains rich rule information as described in Appendix D.

**Metrics:** We use a number of commonly used metrics, including Mean Rank (MR), Mean Reciprocal Rank (MRR), and Hit ratio with cut-off values n = 1,10. MR measures the average rank of all correct entities and MRR is the average inverse rank for correct entities. Hits@n measures the proportion of correct entities in the top n entities. MR is always greater or equal to 1 and the lower MR indicates better performance, while MRR and Hits@n scores always range from 0.0 to 1.0 and

Table 1: Experimental results on FB15k,WN18 and FB15k-R test set. [‡]:E-R(E) denotes EM-RBR(E), indicating that the embedding model in this experimental group is transE. [⋆]:We don't use it here because it's time-consuming and not better than transE on WN18 as reported in Lin et al. (2017).

| Model | FB15k | | | | WN18 | | | | FB15k237 | | | |
|---|---|---|---|---|---|---|---|---|---|---|---|---|
| | MR | MRR | H@1 | H@10 | MR | MRR | H@1 | H@10 | MR | MRR | H@1 | H@10 |
| TransE | 70.3 | 45.77 | 29.98 | 74.27 | 200.9 | 57.47 | 23.21 | 97.68 | 310.2 | 21.57 | 12.45 | 40.4 |
| TransH | 72.56 | 45.81 | 30.37 | 74.01 | 210.7 | 61.94 | 32.03 | 97.49 | 319 | 22.18 | 13.12 | 40.77 |
| TransR | 55.98 | 47.88 | 31.1 | 77.04 | ⋆ - - | - - | - - | - - | 431.3 | 23.49 | 15.35 | 39.96 |
| TransD | 56.41 | 47.88 | 32.48 | 75.99 | 202.8 | 60.35 | 29.6 | 97.37 | - - | - - | - - | - - |
| ‡E-R(E) | 68.36 | 50.01 | 34.44 | 76.23 | **198.1** | **85.23** | 73.94 | **97.83** | **301.9** | 25.26 | 16.5 | 42.83 |
| E-R(H) | 70.72 | 52.39 | **38.82** | 76.52 | 201.4 | 84.57 | 74.97 | 96.48 | 311.8 | **25.58** | 16.81 | **43.09** |
| E-R(R) | 55.47 | 51.93 | 35.86 | **78.35** | ⋆ - - | - - | - - | - - | 422.1 | **25.58** | 17.54 | 41.61 |
| E-R(D) | **55.21** | **53.02** | 38.25 | 78.33 | 201.8 | 84.63 | **75.21** | 97.5 | - - | - - | - - | - - |

higher score reflects better prediction results. We use filtered setting protocol (Bordes et al., 2013), i.e., filtering out any corrupted triples that appear in the KB to avoid possibly flawed evaluation.

**Baseline:** To demonstrate the effectiveness of EM-RBR, we compare with a number of competitive baselines: TransE (Bordes et al., 2013), TransH (Wang et al., 2014), TransR (Lin et al., 2015), TransD (Ji et al., 2015), RUGE (Guo et al., 2017), ComplEx (Trouillon et al., 2016), Dist-Mult (Kadlec et al., 2017) and MINERVA (Das et al., 2017a). Among these state-of-arts, TransE, TransH, TransR and TransD are combined with our reasoning framework. These 8 models are evaluated on 4 standard datasets to prove that our framework is a real reinforced framework. In the end, all the baselines and combined models are evaluated on FB15k-R.

**Implementation:** For TransE, TransH, TransR and TransD, we set the same parameters, i.e., the dimensions of embedding $k = 100$, learning rate $\lambda = 0.001$, the margin $\gamma = 1$. We traverse all the training triplets for 1000 rounds. Other parameters of models are set as the same with the parameters in the published works (Bordes et al., 2013; Wang et al., 2014; Lin et al., 2015)[4]. For RUGE, we set the embedding dimension $k = 100$ and other hyper-parameters are the same with Guo et al. (2017)[5]. For ComplEx and DistMult, all the parameters are consistent with Trouillon et al. (2016)[6]. For MINERVA, we use the implementation of Das et al. (2017a)[7].

## 3.2 EXPERIMENT RESULTS

Firstly, we compare EM-RBR(X) with transX model on FB15k, WN18, FB15k237 and WN18RR. Experimental Results are shown in Table 6. Among the four datasets, the performance on WN18RR is not optimized much cause this dataset is too sparse and its result is shown in E.

When evaluating on FB15k ,WN18 and FB15k-237, our model has improved all the metrics compared with the translation model in the baseline, especially MRR and Hits@1 on each dataset. For example, EM-RBR(D) improve Hits@1 on WN18 from *0.296* to *0.752* compared to transD. While our EM-RBR doesn't have obvious optimization on WN18RR, cause the dataset is too sparse. We have show the experiment result in Appendix E.

Secondly, we compare all the baselines mentioned above on FB15k-R, each triplet in this data set can match a lot of rules so that they can be optimized extremely under EM-RBR. The result is in Table 2. Among those baselines, MINERVA shows the best performance for the test set is rich in rules. While the performance is further optimized compared to MINERVA.

---

[4]https://github.com/thunlp/Fast-TransX
[5]https://github.com/iieir-km/RUGE
[6]https://github.com/ttrouill/complex
[7]https://github.com/shehzaadzd/MINERVA

Table 2: Experimental results on FB15k-R.

| Model | TransE | TransH | TransR | TransD | RotatE | RUGE | ComplEx | DistMult | MINERVA |
|---|---|---|---|---|---|---|---|---|---|
| MRR | 26.11 | 30.43 | 18.51 | 26.16 | 33.29 | 49.14 | 51.1 | 46.2 | 57.3 |
| Hits@1 | 14.9 | 18.25 | 7.65 | 14 | 20.05 | 33.05 | 35.8 | 30.2 | 42.2 |
| Hits@10 | 48.1 | 54.95 | 36.7 | 50.45 | 59.75 | 78.2 | 79 | 77.1 | 84.9 |

| Model | EM-RBR(transE) | EM-RBR(transH) | EM-RBR(transR) | EM-RBR(transD) |
|---|---|---|---|---|
| MRR | 79.88 | 85.61 | **86.01** | 82.04 |
| Hits@1 | 65.1 | **75.45** | 74.3 | 70.45 |
| Hits@10 | 96.4 | 97.8 | **99.2** | 96.15 |

## 3.3 RESULT ANALYSIS

Actually, only a small part of triples in the test set is optimized. In FB15k, the ratio of this optimization is 1/6. Various metrics have been improved much even with this small ratio. The capabilities of our model can be fully demonstrated on the FB15k-R, because each triplet in this set has many rules that can be matched to obtain a good optimization effect. We have recorded each triple's rank under transX and EMRBR(X) respectively as shown in Appendix F.

## 4 RELATED WORK

For the path-based methods, Lao et al. (2011) uses Path Ranking Algorithm (PRA) (Lao & Cohen, 2010) to estimate the probability of an unseen triplet as a combination of weighted random walks. Zhang et al. (2020) and (Qu & Tang, 2019) are both the combination of Markov logic network and embedding. Kok & Domingos (2007) is mainly a clustering algorithm, clustering entity sets under multiple relationship categories. Gardner et al. (2014) makes use of an external text corpus to increase the connectivity of KB. The Neural LP model (Yang et al., 2017) compiles inferential tasks into differentiable numerical matrix sequences. Besides, many studies have modeled the path-finding problem as a Markov decision-making process, such as the DeepPath model (Xiong et al., 2017) and MINERVA (Das et al., 2017b). For the embedding methods, Nguyen (2017) has organized the existing work. Our paper divides all embedding methods into four categories, which are: translation, Bilinear & Tensor, neural network and complex vector. Firstly, for translation, the Unstructured model (Bordes et al., 2014) assumes that the head and tail entity vectors are similar without distinguishing relation types. The Structured Embedding (SE) model (Bordes et al., 2011) assumes that the head and tail entities are similar only in a relation-dependent subspace. Later, there are transE, transR, transH (Bordes et al., 2013; Lin et al., 2015; Wang et al., 2014), etc. Sadeghian et al. (2019) mines first-order logical rules from knowledge graphs and uses those rules to solve KBC. Additionally, other work (Yang et al., 2017; Galárraga et al., 2013) can extract some high-quality rules from knowledge base. For the second type, DISTMULT (Yang et al., 2014) is based on the Bilinear model (Nickel et al., 2011) where each relation is represented by a diagonal matrix rather than a full matrix. SimplE (Kazemi & Poole, 2018) extends CP models (Hitchcock, 1927) to allow two embeddings of each entity to be learned dependently. The third method is to implement embedding with a neural network. Apart from the models mentioned in Section 1, NTN (Socher et al., 2013) and ER-MLP (Dong et al., 2014) also belong to this method. Fourthly, instead of embedding entities and relations in real-valued vector space, ComplEx (Trouillon et al., 2016) is an extension of DISTMULT in the complex vector space. ComplEx-N3 (Lacroix et al., 2018) extends ComplEx with weighted nuclear 3-norm. Also in the complex vector space, RotatE (Sun et al., 2019) defines each relation as a rotation from the head entity to the tail entity. QuatE (Zhang et al., 2019) represents entities by quaternion embeddings (i.e., hypercomplex-valued embeddings) and models relations as rotations in the quaternion space.

## 5 CONCLUSION & FUTURE WORK

This paper introduces an innovative framework called EM-RBR combining embedding and rule-based reasoning, which can be easily integrated with any translation based embedding model. Unlike

previous joint models trying to get better embedding results from rules and triplets, our model allows solving completion from the reasoning perspective by conducting multi-relation path prediction, i.e. a breadth first search. We also demonstrate that EM-RBR can efficiently improve the performance of embedding methods for KGC. This makes the existing translation based embedding methods more suitable and reliable to be used in the real and large scale knowledge inference tasks.

There are two possible directions in the future. On one hand, we will combine our model with more embedding models, not just the translation-based embedding model. On the other hand, we are going to extract more and more reliable association rules to optimize our work. As mentioned above, only a part of triples are optimized when evaluating on FB15k. The fundamental reason for the rest is that there is no corresponding rule for matching. If these two problems are solved, EM-RBR can be better improved.

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

# A Rule's visualization

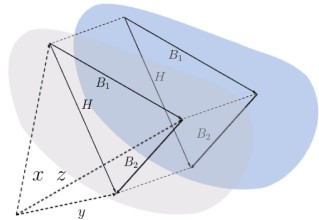

Figure 3: visualization of a rule.

# B Pseudo code

---

**Algorithm 1** EM-RBR

---

**Input:** $(h, r, t)$
**Output:** $\Phi_{\sim(h,r,t)}$
1: Initialize $Q$ as an empty priority queue
2: Initialize the first state: $s_0 \leftarrow \{(h,r,t)\}, \mathcal{H}_{\sim s_0}(h,r,t), \mathcal{L}_{\sim s_0}(h,r,t) \leftarrow 1, s_{\sim transX}(h,r,t)$
3: Initialize the score: $\Phi_{\sim(h,r,t)} \leftarrow s_{\sim transX}(h,r,t)$
4: $Q$.push($s_0$)
5: **while** $!Q$.empty() **do**
6:     $s_{cur} \leftarrow Q$.pop()
7:     $\Phi_{\sim(h,r,t)} \leftarrow \min\{\Phi_{\sim(h,r,t)}, \mathcal{L}_{\sim s_{cur}}(h,r,t)\}$
8:     **if** $\mathcal{H}_{\sim s_{cur}}(h,r,t) < \Phi_{\sim(h,r,t)}$ **then**
9:         **for** each $s_{ne}$ extended from $s_{cur}$ with rule $(B_1, B_2, H)$ **do**
10:             $\mathcal{H}_{\sim s_{ne}}(h,r,t) = \mathcal{H}_{\sim s_{cur}}(h,r,t) * \omega(B_1, B_2, H)$
11:             $\mathcal{L}_{\sim s_{ne}}(h,r,t) = \mathcal{H}_{\sim s_{ne}}(h,r,t) * \prod_{(h',r',t') \in s_{ne}} s_{\sim transX}(h',r',t')$
12:             **if** $\mathcal{H}_{\sim s_{ne}}(h,r,t) < \mathcal{L}_{\sim s_{cur}}(h,r,t)$ **then**
13:                 $Q$.push($s_{ne}$)
14:             **end if**
15:         **end for**
16:     **end if**
17: **end while**

---

# C Data message in example

Table 3: Triplets in each state. [$\star$]: If this symbol appears in the upper right corner of a triple, the triplet is not in the knowledge graph. Other triplets are all in the knowledge graph.

| state | triplets | | | |
|-------|----------|---|---|---|
| $s_1$ | $(h, B_1, m_1)^\star$ | $(m_1, B_2, t)$ | | |
| $s_2$ | $(h, B_3, m_2)$ | $(m_2, B_4, t)$ | | |
| $s_3$ | $(h, B_5, m_3)^\star$ | $(m_3, B_6, m_1)$ | $(m_1, B_2, t)$ | |
| $s_4$ | $(h, B_5, m_4)^\star$ | $(m_4, B_6, m_1)$ | $(m_1, B_2, t)$ | |
| $s_5$ | $(h, B_7, m_5)^\star$ | $(m_5, B_8, m_1)$ | $(m_1, B_2, t)$ | |
| $s_6$ | $(h, B_9, m_6)$ | $(m_6, B_{10}, m_4)^\star$ | $(m_4, B_6, m_1)$ | $(m_1, B_2, t)$ |
| $s_7$ | $(h, B_{11}, m_7)$ | $(m_7, B_{12}, m_4)^\star$ | $(m_4, B_6, m_1)$ | $(m_1, B_2, t)$ |
| $s_8$ | $(h, B_9, m_6)$ | $(m_6, B_{13}, m_8)^\star$ | $(m_8, B_{14}, m_4)$ | $(m_4, B_6, m_1)$ | $(m_1, B_2, t)$ |
| $s_9$ | $(h, B_9, m_6)$ | $(m_6, B_{13}, m_9)$ | $(m_9, B_{14}, m_4)^\star$ | $(m_4, B_6, m_1)$ | $(m_1, B_2, t)$ |

Table 4: Rule's score

| rule | score |
|------|-------|
| $B_1(x,z) \wedge B_2(z,y) \Rightarrow r(x,y)$ | r1 |
| $B_3(x,z) \wedge B_4(z,y) \Rightarrow r(x,y)$ | r2 |
| $B_5(x,z) \wedge B_6(z,y) \Rightarrow B_1(x,y)$ | r3 |
| $B_7(x,z) \wedge B_8(z,y) \Rightarrow B_1(x,y)$ | r4 |
| $B_9(x,z) \wedge B_{10}(z,y) \Rightarrow B_5(x,y)$ | r5 |
| $B_{11}(x,z) \wedge B_{12}(z,y) \Rightarrow B_5(x,y)$ | r6 |
| $B_{13}(x,z) \wedge B_{14}(z,y) \Rightarrow B_{10}(x,y)$ | r7 |

Table 5: Dataset used in our study

| Dataset | Relations | Entities | Train | Validation | Test |
|---------|-----------|----------|-------|------------|------|
| FB15k | 1,345 | 14,951 | 483,142 | 50,000 | 59, 071 |
| WN18 | 18 | 40,943 | 141,442 | 5,000 | 5,000 |
| FB15k237 | 237 | 14,541 | 272,115 | 17,535 | 20, 466 |
| WN18RR | 11 | 40943 | 86,835 | 3,034 | 3,134 |
| FB15k-R | 1,345 | 14,951 | 483,142 | 50,000 | 1000 |

## D  DATASET USED IN OUR EXPERIMENT

Some triples in the test set of FB15k are selected to make up of a new test set named FB15k-R. We construct the new test set with the following steps. Firstly, sort all triples in the test set by some indicator, denoted as $Z(h,r,t)$. Secondly, randomly select triples within different ranking ranges with a certain probability distribution, until 1,000 triples are selected. The probability distribution is described as Equaion 6.

$$
P(event) = \begin{cases}
0.1, & event \leftarrow select\ a\ triple\ whose\ Z = 0 \\
0.2, & event \leftarrow select\ a\ triple\ whose\ ranking \leq 10000 \\
0.3, & event \leftarrow select\ a\ triple\ with\ ranking\ \tau, \tau > 10000\ \&\ \tau \leq 30000 \\
0.4, & event \leftarrow select\ a\ triple\ whose\ ranking \geq 30000\ \&\ Z \neq 0
\end{cases} \quad (6)
$$

$Z(h,r,t)$ can be calculated as one of the candidates in the following.

- The number of rules each triple can match, and the matching method is the same to that in Section 2.2.1.

- The number of valid rules each triple can match. A valid rule meets $\omega < s_{\sim transE}(h,r,t)$, where $\omega$ denotes the score of a rule and $s_{\sim transE}(h,r,t)$ denotes the score of the triple under transE.

- The maximum $s_{\sim transE}(h,r,t) - \omega_{rule}$ for each triple, and $rule \in rule_{(h,r,t)}$. $rule_{(h,r,t)}$ is the set of rules $(h,r,t)$ can match.

- $s_{\sim transE}(h,r,t) - s_{\sim reasoning}(h,r,t)$, where $s_{\sim reasoning}(h,r,t)$ denotes the score of $(h,r,t)$ under EM-RBR(transE).

In our formal experiment, we choose the last strategy to generate FB15k-R.

## E  EXPERIMENT RESULT OF WN18RR

The metrics reported in this table is not better than that reported in other work[8]. The different hyper-parameters result in this. This experiment is a verification of the enhancement effect, so it makes sense only when comparing transX and EM-RBR(X).

---

[8]https://github.com/thunlp/OpenKE

Table 6: Experimental results on WN18RR.

| Model | MR | MRR | H@1 | H@10 |
|---|---|---|---|---|
| TransE | 5338 | 19.92 | 1.04 | 43.43 |
| TransH | 5494 | 19.58 | 2.28 | 42.95 |
| ‡EM-RBR(transE) | 5477 | 19.25 | 1.08 | 44.07 |
| EM-RBR(transH) | 5597 | 19.31 | 1.52 | 43.68 |

# F  SINGLE TRIPLE RANKING ANALYSIS

In order to better understand the specific situation being optimized on each triplet. We respectively analyzed the corresponding ranking of each triplet under the translation model and the EM-RBR model when the head entity replacement and tail entity replacement were performed. The results were displayed in Table 7. The data item in the table is the result of sorting from largest to smallest value of $s_{\sim trans} - s_{\sim ER}$, where $s_{\sim trans}$ is the ranking under the corresponding translation model and $s_{\sim E-R}$ is the ranking under the corresponding EM-RBR model.

Table 7: Optimized case analysis. [✝]: the id number of the test case, for example, the first test case is */m/01qscs /award/award_nominee/award_nominations./award/award_nomination/award /m/02x8n1n* and its id number is 0. [∗]: the rank of the test case in EM-RBR. [‡]: the rank of the test case in the embedding model. [◇]: L corresponds to replacing the head entity and R the tail entity.

| Rank | EM-RBR(**E**) | | | | EM-RBR(**H**) | | | | EM-RBR(**R**) | | | |
|---|---|---|---|---|---|---|---|---|---|---|---|---|
| | ✝id | ∗E-R | ‡trans | ◇L/R | ✝id | ∗E-R | ‡trans | ◇L/R | ✝id | ∗E-R | ‡trans | ◇L/R |
| 1 | 47722 | 2 | 14141 | R | 18355 | 2 | 12689 | L | 15105 | 2 | 966 | L |
| 2 | 47722 | 2 | 13900 | L | 32966 | 3 | 8551 | L | 42675 | 1 | 868 | R |
| 3 | 18355 | 2 | 7525 | L | 18355 | 2 | 7231 | R | 34891 | 2 | 733 | R |
| 4 | 36133 | 2 | 6884 | L | 47722 | 2 | 4569 | L | 24314 | 1 | 714 | R |
| 5 | 33004 | 1 | 6253 | L | 24243 | 1 | 4547 | L | 32849 | 1 | 701 | L |
| 6 | 33243 | 2 | 5883 | R | 33004 | 1 | 4490 | L | 55951 | 2 | 673 | L |
| 7 | 30883 | 2 | 5674 | R | 47722 | 2 | 3977 | R | 38773 | 1 | 640 | L |
| 8 | 14035 | 2 | 4862 | L | 13358 | 5 | 3741 | R | 54283 | 52 | 674 | L |
| 9 | 18355 | 2 | 4525 | R | 55951 | 2 | 3699 | L | 25500 | 1 | 585 | R |
| 10 | 24243 | 1 | 3655 | L | 50019 | 1 | 3386 | R | 34891 | 2 | 555 | L |
| . . . | | | | | | | | | | | | |
| 19372 | 52886 | 4 | 2 | R | 23339 | 6 | 1 | L | 44273 | 2 | 1 | R |
| 19373 | 52707 | 13 | 11 | L | 23288 | 7 | 2 | R | 43969 | 2 | 1 | R |
| 19374 | 52529 | 9 | 7 | R | 23218 | 7 | 2 | R | 43664 | 2 | 1 | L |
| 19375 | 51447 | 3 | 1 | R | 21906 | 7 | 2 | R | 43483 | 2 | 1 | L |
| 19376 | 50932 | 4 | 2 | R | 20794 | 7 | 2 | R | 42380 | 2 | 1 | R |
| . . . | | | | | | | | | | | | |

