# OpenReview forum: "EM-RBR: a reinforced framework for knowledge graph completion from reasoning perspective"
_ICLR.cc/2021/Conference — Reject_

### Official Review · AnonReviewer1 · 2020-10-27
**Not ready for submission. Very difficult to understand and does not compare results to comparative models.**

**Rating:** 3
**Confidence:** 3

**Review:**

Summary: The paper seek to improve KG representation (e.g. for link prediction and question answering) by combining logical reasoning (logical rule templates) with statistical methods (TransE). Rules are mined from the KG using AMIE and recursive backward steps are taken, using the mined rules, to determine if a fact is true.

Quality: the paper is very difficult to read, to an extent that puts it below the quality I would expect for a top conference.

Clarity: As above, the paper is very difficult to read, and even pushing through grammatical errors, the algorithm itself is very difficult to understand. In many cases, terms are used first and defined later, making it hard to comprehend and requiring a lot of going back-and-forth. As one specific example, shortly after definition 2.2, H_{s_cur} is compared to \phi, which hasn't been given a value (the clearest explanation is given in an appendix - in my view this should be in the main body since it essential). A footnote says that will be initialised to L_{s_o}, but that value (including the meaning of s_o) is not yet defined. As another example, \Delta_path appears in Eq 1, having not been previously defined (including any notion of "path"), it is then defined loosely in terms of "rules used in the extension ...", but this is fundamental to the algorithm and should be stated concretely (perhaps built iteratively with each extension step).

Originality: The model uses a pre-existing rule mining algorithm (AMIE) and KG model (TransE) as a basic triplet score function within an exhaustive logical search framework. Using BFS to find supporting predicates does not seem particularly novel, but the means of combining score functions seems to be.

Significance: Hard to say given the difficulty understanding the paper overall and its apparent unbounded complexity.

Pros: The subject tackled, i.e. combining logical reasoning and statistical learning, is an important area of research.

Cons:
- The paper is insufficiently clear and the poor grammar makes it too difficult to comprehend to be acceptable at a top conference.
- Many other works combine logical rules and KG embeddings but are not compared to. It is unclear why not. The KG embedding models (that do not use logical rules) compared to are not representative of state-of-art, e.g.
  RotatE (Zhiqing Sun, Zhi-Hong Deng, Jian-Yun Nie, and Jian Tang. Rotate: Knowledge graph embedding by relational rotation in complex space. arXiv preprint arXiv:1902.10197, 2019.)
  QuatE (Shuai Zhang, Yi Tay, Lina Yao, and Qi Liu. Quaternion knowledge graph embeddings. In Advances in Neural Information Processing Systems, pp. 2731–2741, 2019)
  Tucker (Ivana Balaˇzevi´c, Carl Allen, and Timothy M Hospedales. Tucker: Tensor factorization for knowledge graph completion. arXiv preprint arXiv:1901.09590, 2019.)
  MuRP (Ivana Balazevic, Carl Allen, and Timothy Hospedales. Multi-relational poincar´e graph embeddings. In Advances in Neural Information Processing Systems, pp. 4465–4475, 2019.)
- The approach of breadth first search has the potential for exponential explosion and seems something that embedding approaches avoid by capturing these relationships implicitly. In that respect the proposed approach appears retrograde.

---

### Official Review · AnonReviewer2 · 2020-10-28
**Combination of rule learning and embedding learning for knowledge reasoning**

**Rating:** 4
**Confidence:** 4

**Review:**

The work utilizes relational background knowledge contained in logical rules to conduct multi-relational reasoning for knowledge graph (KG) completion. This is different from the superficial vector triangle linkage used in embedding models. It solves the KG completion task through rule-based reasoning rather than using rules to obtain better embeddings. Experiments on FB15K, WN18, and a new dataset FB15K-R demonstrate the effectiveness of the proposed model EM-RBR.

The work is well motivated. As for KG completion, embedding-based methods usually fail to predict long-tail relations, while rules can alleviate the problem. Hence, the work proposes to use a rating mechanism combined with a rule-based reasoning process for ranking test triplets. Generally, the technique sounds reasonable.

A minor question:

1) In Algorithm 1, why to push the state into the queue when H_snew < L_scur? My understanding is that if H_snew >= L_scur, and we already know L_snew >= H_snew, then we can get L_snew >= L_scur. In this case, it is not necessary to push the state into the queue. Right? I think the authors should further clarify this.

Some suggestions:

1) The paper demonstrates that the proposed EM-RBR(X) performs better than the X model, but I think it is better to compare with more models that utilize rules for KG completion.

2) FB15k and WN18 are too old and suffer from the test data leakage issue. Recent studies for KGC no longer use FB15k and WN18. I would suggest the authors to try FB15k-237 and WN18RR. Besides, I found that if many of the test triples can not be matched by rules, the performance improvement would be subtle. I think in sparse KGs like WN18RR or FB15k-237, the proposed model will also suffer this problem, as the rules mined by AMIE will also be sparse.

3) It is better to provide a rule-based reasoning path that can explain why the inferred triple is true. I think such a case study is meaningful.

Minor mistakes (my point of view):

1) On page 3, it says that "The initial state is the target triplet itself, whose H is 1 and L is its score under an embedding model", and at the footnote of page 3, it says that "Φ is defined as Definition 2.2 and initialized as L_{s0}". However, Algorithm 1 of Appendix B initializes H and L as 1. And in section 2.3 (page 4), it says that "The initial state s0 only contains one triplet (h, r, t), and its state score and heuristic score are both 1". I think it is very confusing.

2) On page 4 (Section 2.2.2), it says that  "kB1 + B2 − Hk ∈ R 3*k". I think it may be wrong.

3) I think Algorithm 1 of Appendix B lacks the procedure to handle the state whose triples are all in the KG. Just like in Figure 2 (I), I don't know which part in Algorithm 1 is used to update \phi using L_s2.

4) In Table 3, the s3 state is {(h, B5, m3) (m3, B6, m1)* (m1, B2, t)}. And in Table 5, the rule 5 (r5) is "B9(x, z) ∧B10(z, y) ⇒ B5(x, y) r5". Based on these, I think the example extension in Figure 2 (III) may be wrong.

Overall, I think this paper is interesting, but it needs further improvement.

-- after rebuttal --

Thank the authors for their responses.

---

### Official Review · AnonReviewer3 · 2020-10-28
**Official Blind Review #3**

**Rating:** 4
**Confidence:** 4

**Review:**

Summary:

This paper proposes a novel algorithm for the task of Knowledge Graph Completion. Proposed algorithm uses reasoning rules from Knowledge base jointly with Translation Embedding models to gather rich information for multi-relation reasoning link prediction in Knowledge Graphs. This enables the algorithm to explore relation between two entities in a deeper context.

-------------------------------------------------------------------------------------------------------------------------------------------------------------------------------------

Pros:

+ Personally, I feel like the problem tackled in this paper is highly practical. Proposed algorithm proves the importance of high quality rules in a Knowledge Base. This can motivate further research in constructing new Knowledge Bases with rules rich relations or mining high quality rules from already existing Knowledge Bases.

+ This paper also introduces a new rules rich dataset named FB15k-R. The dataset can be used in future work to ascertain the quality of rule mining and for judging performance of methods on Link Prediction task as well.

+ The proposed algorithm is novel and achieves SOTA performance on variety of Knowledge Graph Link Prediction datasets.

-------------------------------------------------------------------------------------------------------------------------------------------------------------------------------------

Cons:

- Overall the paper is well written but some details seem to be missing or are not consistent (see questions below).

- The motivation for choosing this algorithm is not explained in the paper. This makes the algorithm a little hard to understand.

- Very minimal details are provided about the new dataset FB15k-R. The paper should include the dataset generation strategy/algorithm (at least in the appendix section).

-------------------------------------------------------------------------------------------------------------------------------------------------------------------------------------

Overall, I vote for accepting the paper. This paper can certainly be improved in a number of ways but the contributions are worthwhile and it may inspire some interesting future work. Hopefully the authors can address my concern in the rebuttal period.

-------------------------------------------------------------------------------------------------------------------------------------------------------------------------------------

Questions:

- How does equation 1 change in case of rules like B(x,y) => H(x,y)?

- Both heuristic score and phi are initialised with 1 in the starting. The stopping criteria for the algorithm is written as H_scur >= phi (section 2.2). How will the algorithm start with this criteria? Is this a mistake or am I missing something here?

- In Definition 2.1, what do you mean by "where the entities order in one triplet is random, i.e. B3(z; x)^B4(z; y) =>
R(x; y) is also a valid rule."? This statement seems to imply that order of the entities doesn't matter, but in reality it should matter. Maybe the statement can be rephrased?

- [AMIE](http://resources.mpi-inf.mpg.de/yago-naga/amie/amie.pdf) is used for auto-mining rules from the KB. There are several other method like [DRUM](https://papers.nips.cc/paper/9669-drum-end-to-end-differentiable-rule-mining-on-knowledge-graphs.pdf), [Neural-LP](https://arxiv.org/pdf/1702.08367.pdf) for auto-mining of rules. Were they tried before deciding on AMIE?

- In example section, the algorithm tries to calculate the score of the target triplet by going one step one level deeper (expanding the current triplet). The process resembles to DFS on the graph structure shown in Figure 2.
Earlier it is mentioned that the algorithm is based on BFS and it is attributes to the use of Priority Queue. For me this creates inconsistency while understanding the algorithm. Could you please provide an example of how it resembles BFS?

- While ranking all the possible triplets, how is a tie between scores resolved?

-------------------------------------------------------------------------------------------------------------------------------------------------------------------------------------

Minor Comments/Typos:

- Section 2.2: states that were **pop** from Q -> states that were **popped** from Q
- Footnote 2: B(x,z) => H(x,y) -> B(x,y) => H(x,y)

Post Rebuttal Comments: Following the discussion in rebuttal phase and after reading all the other reviews (and authors response) , I feel that the paper is not ready for submission. While authors did address some of my concerns, the evaluation strategy seems flawed and the compared methods are not representative (as pointed out by reviewer 1). As a result, I am changing my rating from 6 to 4. Thanks!

---

### Official Review · AnonReviewer4 · 2020-10-28
**Method not clearly explained, problems in evaluation**

**Rating:** 3
**Confidence:** 4

**Review:**

The paper proposes a framework (EM-RBR) for doing Knowledge Base (KB) completion. Instead of the direct triple score from an embedding based method, EM-RBR allows the triple score to be calculated as a composition of the scores of the rules mined from the KB. EM-RBR uses a BFS type algorithm that recursively searches for reasoning paths connecting the triple while also updating the score.  The authors show that EM-RBR when used as an addendum to a translation-based embedding method (such as TrasnE, TransH) is able to outperform them. They show their results on FB15k and WN18.

Issues:
- The method section is not clearly explained and it was hard to understand the algorithm
- The algorithm performs BFS on every example which is bound to increase the inference time, the authors claim that the approach is efficient and so should provide some analysis on that
- The paper talks about other path-/rule-based methods, the authors should compare their approach with them (like Neural LP, MINERVA, DeepPath, etc). (Approaches that combine embedding based and rule-based methods are not novel, example: https://arxiv.org/pdf/1506.00379)
- The authors evaluate their work on FB15k and WN18, which have been shown to have several issues (leaking of test triples as inverses, etc). Some of the current benchmarks for the task are FB15k-237 (https://www.aclweb.org/anthology/W15-4007.pdf) and WN18RR (https://arxiv.org/abs/1707.01476).
- The authors show results on a version of FB15k they create called FB15k-R. The only thing they talk about it is
  >  We create FB15k-R, a subset of FB15k, which contains 1000 tested
triplets that have rich rules to take reasoning"

- **This is a major problem**. There is no explanation regarding how this subset was chosen. It seems that it was chosen specifically because their model performed well on it. This is blatant cherry-picking.


On the basis of the above points, I recommend rejection.

---

### Decision · Program_Chairs · 2021-01-07
**Final Decision**

**Decision:**

Reject

**Comment:**

The paper combines logical reasoning and statistical methods to improve knowledge graph completion. Rules are mined from the KG using AMIE and recursive backward steps are taken, using the mined rules, to determine if a fact is true. The reviewers agree that the paper can be improved by explaining more details of the method to make it more easy to understand.